# Early onset of neurological features differentiates two outbreaks of Lassa fever in Ebonyi state, Nigeria during 2017–2018

Nneka M. Chika-Igwenyi[1]*, Rebecca E. Harrison[2], Christina Psarra[3], Julita Gil-Cuesta[3], Maria Gulamhusein[3], Emeka O. Onwe[4], Robinson C. Onoh[5], Uche S. Unigwe[1,6], Nnennaya A. Ajayi[1], Ugochukwu U. Nnadozie[7], Chiedozie K. Ojide[8,9], Damian U. Nwidi[1], Obumneme Ezeanosike[4], Emeka Sampson[10], Azuka S. Adeke[11], Collins N. Ugwu[1], Uchenna Anebonam[12], Jacques K. Tshiang[13], Jacob Maikere[13], Anthony Reid[3]

1 Department of Medicine, Alex Ekwueme Federal University Teaching Hospital Abakaliki, Abakaliki, Ebonyi, Nigeria, 2 Médecins sans Frontières, Operational Centre Brussels, Belgium, 3 Médecins sans Frontières Operational Research Unit (LuxOR), Operational Centre Brussels, Belgium, 4 Department of Paediatrics, Alex Ekwueme Federal University Teaching Hospital Abakaliki, Abakaliki, Ebonyi, Nigeria, 5 Department of Obstetrics and Gynaecology, Alex Ekwueme Federal University Teaching Hospital Abakaliki, Abakaliki, Ebonyi, Nigeria, 6 University of Nigeria Teaching Hospital, Ituku Ozalla Enugu, Nigeria, 7 Division of Plastic Surgery, Department of Surgery, Alex Ekwueme Federal University Teaching Hospital Abakaliki, Abakaliki, Ebonyi, Nigeria, 8 Department of Medical Microbiology, Alex Ekwueme Federal University Teaching Hospital Abakaliki, Abakaliki, Ebonyi, Nigeria, 9 Virology Laboratory, Virology Centre,AlexEkwueme Federal University Teaching Hospital Abakaliki, Abakaliki, Ebonyi, Nigeria, 10 Ebonyi State Ministry of Health, Abakaliki, Ebonyi, Nigeria, 11 Department of Community Medicine, Alex Ekwueme Federal University Teaching Hospital Abakiliki, Abakaliki, Ebonyi, Nigeria, 12 World Health Organization, Abuja, Nigeria, 13 Médecins sans Frontières, Abuja, Nigeria

* nnekaigwenyi@gmail.com

## Abstract

Lassa fever (LF) is an acute viral haemorrhagic illness with various non-specific clinical manifestations. Neurological symptoms are rare at the early stage of the disease, but may be seen in late stages, in severely ill patients.The aim of this study was to describe the epidemiological evolution, socio-demographic profiles, clinical characteristics, and outcomes of patients seen during two Lassa fever outbreaks in Ebonyi State, between December 2017 and December 2018.

Routinely collected clinical data from all patients admitted to the Virology Centre of the hospital during the period were analysed retrospectively. Out of a total of 83 cases, 70 (84.3%) were RT-PCR confirmed while 13 (15.7%) were probable cases. Sixty-nine (83.1%) patients were seen in outbreak 1 of whom 53.6% were urban residents, while 19%, 15%, and 10% were farmers, students and health workers respectively. There were 14 (16.8%) patients, seen in second outbreak with 92.9% rural residents. There were differences in clinical symptoms, signs and laboratory findings between the two outbreaks. The case fatality rates were 29.9% in outbreak 1 and 85.7% for outbreak 2. Neurological features and abnormal laboratory test results were associated with higher mortality rate, seen in outbreak 2. This study revealed significant differences between the two outbreaks. Of particular concern was the higher case fatality during the outbreak 2 which may be from a more

**Data Availability Statement:** All relevant data are within the manuscript and its Supporting Information files.

**Funding:** This work was supported by funds from the United Kingdom's Department for International Development (DFID, Grant ref: 202506-103) and La Fondation Veuve Emile Metz-Tesch. The funders had no role in study design, data collection and analysis, decision to publish, or preparation of the manuscript.

**Competing interests:** The authors have declared that no competing interests exist.

virulent strain of the Lassa virus. This has important public health implications and further molecular studies are needed to better define its characteristics.

## Author summary

Neurological manifestations are uncommon in the early stages of Lassa Fever. In Ebonyi State, an unusual pattern was observed between two outbreaks that occurred quite close together in time but with distinctly different presentations. Previous studies on the 2018 outbreak focused on the first part of the outbreak between December 2017—May 2018. We therefore compared the two outbreaks and observed that patients in the second outbreak which occurred between August 2018-December 2018, presented with early neurological symptoms and a high mortality rate. This observation highlights the need for further evaluation and molecular studies.

## Introduction

Lassa fever (LF) is a viral haemorrhagic illness caused by Lassa virus (LASV); associated with frequent fatal outcomes, and is endemic in West Africa [1–3].

LASV is mainly transmitted to humans through exposure to infected rodent urine, faeces, tissue or blood [2] or through direct contact with urine, blood, respiratory secretions or other body fluids of an infected person [4,5]. Hospital-acquired (nosocomial) transmission from person to person can occur if appropriate Personal Protective Equipment (PPE) is not used in management of suspected cases [6–8]. LF symptoms may be mild or asymptomatic in about 80% of cases, but can otherwise cause acute illness [9,10]. Clinical disease usually begins within the first three weeks [11] after exposure with flu-like illness, characterized by a generalized weakness, malaise and fever which may be accompanied by a wide spectrum of non-specific clinical manifestations, at different stages of the disease [12]. Neurological manifestation (Tremors, Seizures, Disorientation, Coma etc) are uncommon in early disease, but may be seen in later stages [4,13,14]. Some studies suggest that early manifestation of central nervous system features could reflect a poor prognosis [11,15]. In severe disease, death may occur within 14 days of onset [4,16]. The clinical course of the disease is demonstrated in (S1 Table) [10].

An estimated number of 100,000–300,000 people are infected with LASV every year in West Africa with approximately 5,000 deaths annually [12,17,18]. The disease is one of the public health risk in countries like Sierra Leone, Liberia, Guinea and Nigeria where it is considered endemic and other West African countries outside the endemic zone[19].

The virus is most diverse in Nigeria, with frequent outbreaks and limited scientific evidence to inform medical and public health management of the disease [20–22].

In 2018, there was an unusual increase of LF cases in Nigeria [23,24] and the World Health Organization (WHO) recognized LF as one of the disease with a potential to cause a public health emergency [25]. The Nigeria Centre for Disease Control (NCDC) activated an Emergency Operations Centre (EOC) to coordinate response activities [26] and by December 2018 (week 52), there were reported 633 laboratory-confirmed cases with an estimated case fatality rate (CFR) of 27% [27]. Confirmed cases were concentrated in the south-western states of Edo (44%) and Ondo (25%) while Ebonyi in the south-east contributed (11%) [27].

In Ebonyi state, an unusual pattern was observed in the two outbreaks occurring during the period of December 2017 to December 2018. The first outbreak followed the traditional pattern but the second outbreak presented with more neurological signs and symptoms and resulted in a very high mortality rate. This latter pattern had not been observed before in our environment and was a potential threat and burden to existing LF control activities.

Given the existing knowledge gap on LF [28,29] and the increased need for evidence to inform early diagnosis and treatment, we decided to compare the presenting clinical features, hospital management and the case fatality rates of these two LF outbreaks observed in Ebonyi State. Previous studies on the 2018 outbreak captured mainly the first part between December 2017 - May 2018 [23,30,31] hence, the added value of this study is the addition of a complete and unique data set of cases seen in the second epidemics during this period. In two separate outbreaks of LF in Ebonyi State, this study describes 1) the socio-demographic profiles, epidemiological evolution of the outbreaks and case fatality rates, 2) the clinical characteristics, hospital management and outcomes of the confirmed and probable cases and 3) the clinical symptoms significantly associated with mortality in the two outbreaks.

## Methods

### Ethics statement

The study received ethical approval from AEFUTHA Research and Ethics Committee (approval number FETHA/REC/vol.2/2019/201) and this research fulfilled the exemption criteria by Médecins sans Frontières Ethics Review Board (MSF ERB) for a posteriori analysis of routinely collected clinical data and thus did not require MSF ERB review. It was conducted with permission from the Medical Director, Operational Centre Brussels Médecins sans Frontières.

### Design

A retrospective descriptive study of routinely collected clinical data.

### General setting

Nigeria is a country located on the Western coast of Africa with an estimated population of over 200.96 million and ranking 7th in the world population rating as of 2019 [32,33]. Ebonyi State is in the south-eastern geopolitical zone of Nigeria (Fig 1) with 13 local government areas (Fig 2) and an estimated population of 3.05 million [34]. About 80% of the population are subsistence farmers [35]. The rest are civil servants, bankers and traders. The traders consist of adults and youths in low- and middle-income social classes who engage in wholesale and retail buying/selling of beverages, food stuff, electronics, stationary and textile materials. Alex Ekwueme Federal University Teaching Hospital Abakaliki (AEFUTHA) is located in Abakaliki, the administrative town of Ebonyi State.

### Specific study setting

The study was carried out at the Virology Centre, a dedicated LF Diagnostic and Treatment Centre of AEFUTHA, Ebonyi State in the south-eastern geopolitical zone of Nigeria. AEFUTHA has 720 beds and provides services in major clinical areas of Internal Medicine, Surgery, Paediatrics, Obstetrics and Gynaecology, Family and Community Medicine and other sub specialties. Its staff exceeds 4,000 and it is the only tertiary referral centre in the south-eastern region of Nigeria receiving patients with LF from neighbouring states including Abia, Enugu, Akwa Ibom and Cross River.

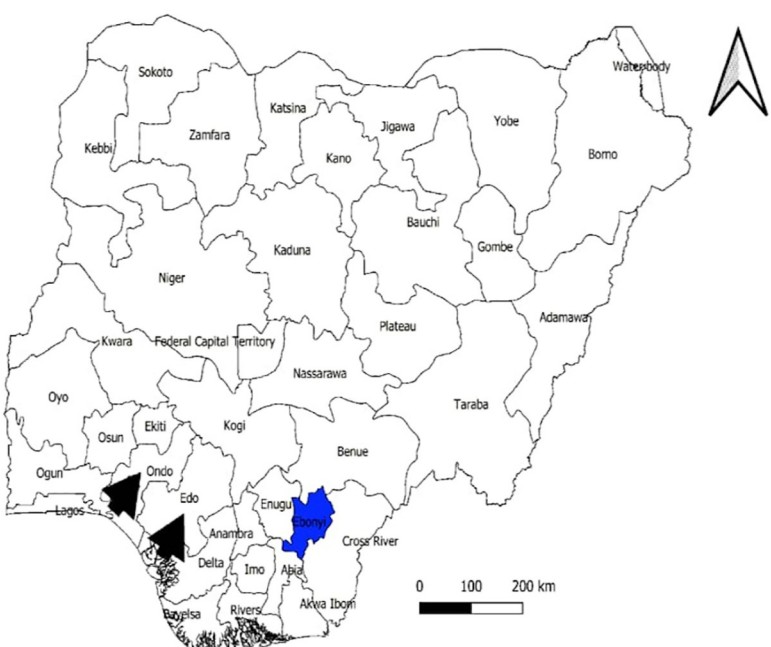

**Fig 1. Map of Nigeria Highlighting Ebonyi State.** With black arrowheads showing other areas mentioned in this study. Developed using ArcGIS software.

The Virology Centre is made up of a 27-bedded ward complex, call rooms and offices, a dialysis unit, waste incinerator, mini morgue (a miniature mortuary outside the ward, where corpses are kept temporarily while arrangements are made for a safe burial), pharmacy, functional laboratory, with over 50 dedicated staff.

A major partner is the NCDC which offers support to its leadership by setting of guidelines, standards and monitoring of activities. Also, the international non-governmental organization, Médecins sans Frontières (MSF), helps with training, while offering human resources and materials. Other partners include WHO, Ebonyi State Ministry of Health, One Health and Accelerating Vaccines for Ebola and the Lassa Fever (OVEL) Project, (Redeemer's University/ Cambridge University group) and Nagasaki University Institute of Tropical Medicine.

## Methodology for LASV sample handling

3 - 5 mls of blood sample was collected from each suspected Lassa patient in EDTA bottle and transported to the Virology laboratory in a cold chain in triple packaged condition in line with NCDC guideline. The plasma was harvested from each sample into a plain bottle in a glove box. The plasma samples were inactivated and RNA extracted through spin column method using QIAamp RNA Mini Kit (www.qiagen.com) in strict compliance with the manufacturer's instruction. Sample inactivation was done inside a glove box while the buffer washings and elution were done on the bench. Altona 2.0 Lassa diagnostic kits that targets the S-gene (GPC) and L-gene of Lassa virus was used for amplification using MIC qPCR cycler. Thermo-cycling conditions were set at 55˚C for 20 minutes (Reverse Transcription), 95˚C for 2 minutes (Denaturation) and 45 amplification cycles at 95˚C for 15, 55˚C for 45 sec and 72˚C for 15 seconds sequentially. Cycle threshold (CT) value was set at 40. Detection of both genes was interpreted as Lassa PCR positive result if negative test control (NTC) and positive test control (PTC) are negative and positive respectively, and internal controls are in order.

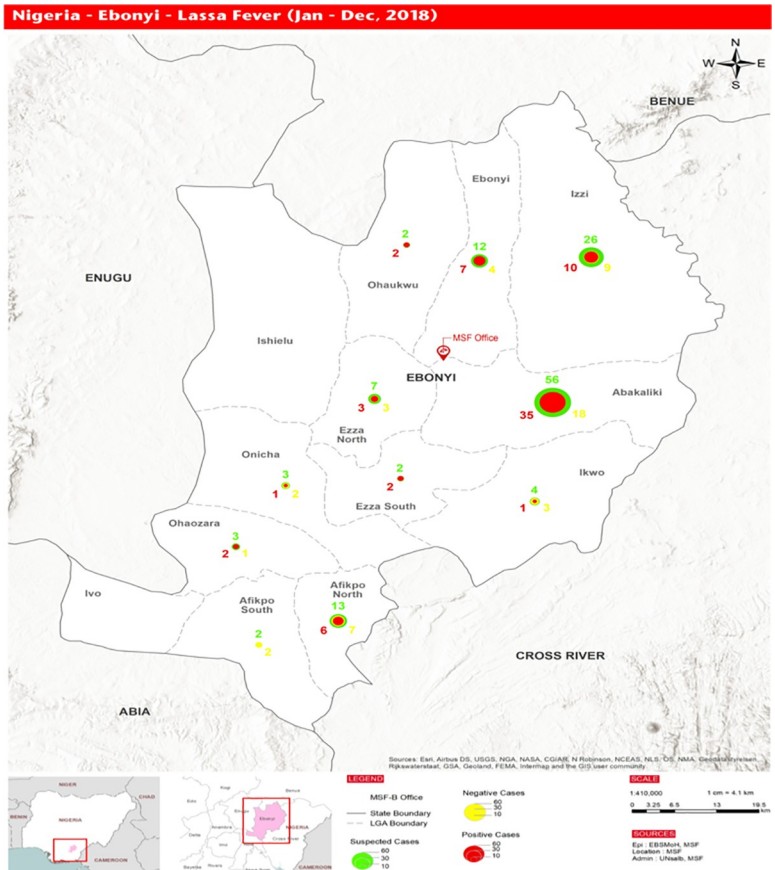

**Fig 2. Map of Ebonyi State showing LGA'S affected by LF in 2018.** Generated by Medécins sans Frontières, (MSF) Nigeria.

## Lassa fever management protocol

Patients are routinely seen at the various entry points of the hospital (Medical and Surgical Emergency, Obstetrics and Gynaecology, Child Emergency). Once they are suspected of having LF and meets the NCDC case definitions criteria (S2 Table), they are isolated in a Holding Area, currently an eight-bedded observation bay, for 24-48 hours while awaiting results of the initial baseline investigations and Lassa virus specific real time Reverse Transcriptase-Polymerase Chain Reaction (RT-PCR) for LF virus. Once the diagnosis is confirmed positive, patients are transferred to the Virology Centre high risk zone for definitive management.

**Suspected case** is a person with one or more of the following: malaise, fever, headache, sore throat, cough, nausea, vomiting, diarrhea, myalgia (muscle pain), central chest pain or retrosternal pain, hearing loss and either a history of contact with rodent excreta, history of contact with a probable or confirmed Lassa fever case within a period of 21 days of onset of symptoms or any person with inexplicable bleeding / hemorrhaging [36].

**Probable case** is any suspected case as defined above but who died without collection of specimens for laboratory testing [36].

**Confirmed case** Any suspected case with laboratory confirmation (positive IgM antibody, PCR or virus isolation) [36]

The LF management protocol is demonstrated in the flow chart (Fig 3).

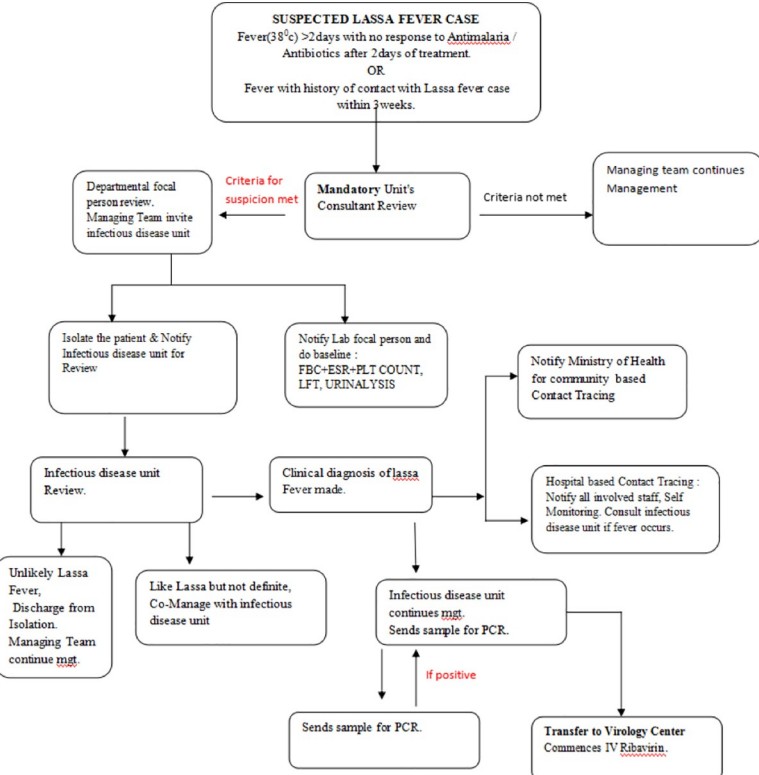

**Fig 3. (Flow chart for the management of suspected Lassa fever cases in AEFUTHA).**

There is currently no approved vaccine that protects against LF [4]. The drug of choice for treatment of LF infection is parenteral Ribavirin [29,36,37]. It is administered intravenously over a period of 10 days as described in S3 Table (McCormick regimen) [29]. The efficacy is said to be uncertain, however outcomes are reported to be more favourable if treatment is commenced within six days of onset of symptoms [36,38].

Management also entails administration of supportive therapy, including intravenous fluids and electrolytes, antibiotics, treatment of intercurrent infections and organ system dysfunction (e.g. acute kidney injury, seizures, and anaemia), close monitoring and regular reviews of clinical status by well trained, skilled, dedicated clinical staff, headed by infectious diseases physicians. Other specialist physicians are invited to handle system specific complications such as kidney failure as needed. RT-PCR is repeated after five and 10 days of Ribavirin therapy. If the test is negative, with improved clinical symptoms, patients are discharged. However, if it remains positive after the 10th day, further treatment is dependent on the clinical state of the patient.

## Study population and period

The study population were all patients admitted to the Virology Centre of AEFUTHA from December 2017 to December 2018.

## Data variables, collection, and sources

Study data was extracted from the clinical records and the line lists of all LF cases managed at the Virology Centre of AEFUTHA from December 2017 until December 2018.

Clinical features on arrival and during treatment were registered by the principal investigator.

The data were stored both in electronic (Excel based) and paper records. Variables collected included registration number, age, sex, occupation, location, date of symptom onset, admission date, treatment date, discharge date, clinical symptoms and signs and laboratory parameters. Data were single-entered from the electronic and paper-based records into an EPI proforma. Cross-checking was done for all records entered.

### Data analysis

Data were imported into Epi Data v 1.2.2 for cleaning and analysis. Analytic statistics were calculated. The number and proportion of cases for categorical variables and means, standard deviations or medians and interquartile ranges (IQR) were presented where appropriate. Chi squared tests were calculated for categorical variables, and associations with mortality were presented with risk ratios, including 95% Confidence Intervals. T-tests and Kruskal-Wallis tests were calculated to compare means and medians respectively. P values were presented for all statistical comparisons as appropriate. An epi curve was generated using Excel. Differences at the 5% level ($p < 0.05$) were regarded as significant.

## Results

The total number of patients diagnosed with LF in the two outbreaks in Ebonyi state from December 2017 to December 2018 was 83; 70 (84.3%) were RT-PCR confirmed and 13 (15.7%) were probable cases but only 64 were actually treated in the Virology Centre; the other 19 died before treatment at the Centre could be commenced. Of these, 69 patients were in outbreak 1 and 14 in outbreak 2.

The socio-demographic profile of confirmed and probable cases in the two outbreaks is shown in **Table 1**. There were slightly more males in Outbreak 1, while patients in Outbreak 2 were overwhelmingly from rural locations. Traders were most common in outbreak 1 while farmers were the majority in outbreak 2. The median time between symptom onset and admission was 8 (IQR: 3–11) for traders and 7 (IQR: 6–14) for farmers. Farmers in outbreak 1 had a median time between onset of symptoms of presentation of 6.5 (IQR:5–14.5) whereas for farmers in outbreak 2 this was 12 (IQR:7–12) but data was missing for 2 patients. Obviously, farmers from the rural setting, showed up later to the hospital than those from the urban probably due to socio-cultural factors and transportation challenges. Also, there is improved public health campaign and surveillance in the urban area.

The clinical characteristics and outcomes of confirmed and probable cases are shown in **Table 2**. outbreak 1 had a much higher proportion of confirmed cases. There were marked differences in clinical and laboratory features between the two outbreaks with neurological features much more common in outbreak 2 at 93% versus 43% in outbreak 1 (p = 0.001). Abnormal laboratory findings identified include elevated serum creatinine, AST, ALT and thrombocytopenia. These has been demonstrated as predictors of mortality in previous studies [21]. Other laboratory parameters include urinalysis with blood and protein deposits, although it did not differ between the two outbreaks.

A comparison of probable and confirmed cases for each outbreak are presented in **Table 3**. For all probable cases an epidemiological link was established. Time from symptom onset to admission were greater for probable cases, and all probably cases were dead at admission, which is why they were not confirmed. Symptom presentation between probable and confirmed cases were similar within outbreaks.

**Table 1.  Socio-demographic profile of confirmed and probable cases in the two outbreaks.**

| Characteristics | | Outbreak 1 Dec. 2017 - Apr. 2018 | | Outbreak 2 Aug. 2018 - Dec. 2018 | | P value |
|---|---|---|---|---|---|---|
| | | n | % | n | % | |
| | Total | 69 | 100.0 | 14 | 100.0 | |
| Age (years) | Mean (SD) | 36.3 (19.4) | | 35.6 (19.2) | | 0.728* |
| Sex | Male | 38 | 55.1 | 7 | 50.0 | 0.728** |
| | Female | 31 | 44.9 | 7 | 50.0 | |
| Location | Urban area | 37 | 53.6 | 1 | 7.1 | <0.001** |
| | Rural area | 32 | 46.3 | 13 | 92.9 | |
| Occupation[a] | Trader | 19 | 29.2 | | 0.0 | 0.008** |
| | Student/Children* | 15 | 23.1 | 6 | 42.9 | |
| | Health worker | 12 | 18.5 | | 0.0 | |
| | Civil Servant | 10 | 15.4 | 1 | 7.1 | |
| | Farmer | 8 | 12.3 | 7 | 50.0 | |
| | Not recorded | 1 | 1.5 | | 0.0 | |
| Hospitalised | Yes | 62 | 89.9 | 9 | 64.3 | 0.013** |
| | No | 7 | 10.1 | 5 | 35.7 | |
| Epidemiological link established | Yes | 19 | 27.5 | 12 | 85.7 | 0.001** |
| | No | 50 | 72.5 | 1 | 7.1 | |
| Time between onset of symptoms and admission | Median (IQR) | 7 (4–13) N = 67 | | 10 (7–12) N = 9 | | 0.278*** |

SD = standard deviation; IQR, interquartile range *students and children aged 0-18years

[a]occupation in outbreak1, n = 65

* T-test

** Chi squared test

*** Kruskal wallis test.

The weekly epidemiologic curve of confirmed and probable LF cases by Outbreak is shown in **Fig 4**. The evolution of the two Outbreaks occurred at separate time periods, with peak incidence in epi week 8 (February) for Outbreak 1 and epi week 45 (October) for outbreak 2.

The clinical features and laboratory findings associated with mortality are shown in **S4 Table**. The risk of mortality in the two outbreaks is distinctly different with a fatality of 29.9% in outbreak 1compared to 85.7% for outbreak 2 (p<0.001). The presence of neurological features as well as abnormal laboratory tests were associated with higher mortality, which were more common in outbreak 2 than in outbreak 1. Patients with neurological features, had a Relative Risk (RR) of dying of 8.5 compared to those without. No demographic variables were significantly associated with mortality: They are not displayed in the table.

The odds and adjusted odds of mortality for a selection of variables that were considered important potential confounders, notably, days to presentation, occupation and location are demonstrated in **Table 4**. The presence of neurological symptoms was included, in one version of the regression only, as it was considered to be on the causal pathway between the changes in course, in Outbreak 2 and the increase in mortality. The odds ratio in mortality of Outbreak 2 compared to Outbreak 1 was 14.1 (p = 0.001), and after adjustment for potential confounders (excluding neurological symptoms), the adjusted odds ratio (AOR) was 10.1 (p = 0.007). When the presence of neurological symptoms was included in the regression, the AOR dropped to 6.6 for Outbreak 2 (p = 0.074) indicating that the presence of neurological symptoms was an important variable explaining the change in mortality between two outbreaks.

**Table 2. Clinical characteristics, hospital management and outcomes of confirmed and probable cases of the two outbreaks.**

| Characteristics | | Outbreak 1 Dec, 2017 –Apr, 2018 | | Outbreak 2 Aug, 2018 –Dec, 2018 | | P value |
|---|---|---|---|---|---|---|
| | | n | % | n | % | |
| | Total | 69 | 100.0 | 14 | 100.0 | |
| Case type | Confirmed | 63 | 91.3 | 7 | 50.0 | <0.001** |
| | Probable | 6 | 8.7 | 7 | 50.0 | |
| Presenting symptoms | | | | | | |
| General | Fever | 60 | 87.0 | 14 | 100.0 | 0.359** |
| | Fatigue | 58 | 84.1 | 14 | 100.0 | 0.276** |
| | Respiratory distress | 53 | 76.8 | 8 | 57.1 | <0.001** |
| | Myalgia | 47 | 68.1 | 9 | 64.3 | 0.022** |
| | Bleeding | 32 | 46.4 | 12 | 85.7 | 0.026** |
| | Sore throat | 32 | 46.4 | 11 | 78.6 | 0.010** |
| | Vomiting | 32 | 46.4 | 11 | 78.6 | 0.095** |
| | Diarrhoea | 32 | 46.4 | 8 | 57.1 | 0.660** |
| | Oedema | 22 | 31.9 | 7 | 50.0 | <0.001** |
| | Hypotension | 21 | 30.4 | 5 | 35.7 | 0.013** |
| | Red eye* | 20 | 29.0 | 11 | 78.6 | <0.001** |
| Neurological | Altered consciousness | 20 | 29.0 | 11 | 78.6 | 0.002** |
| | Seizure | 18 | 26.1 | 13 | 92.9 | <0.001** |
| | Neck pain | 8 | 11.6 | 10 | 71.4 | <0.001** |
| | Hearing Impairment | 4 | 5.8 | 4 | 28.6 | <0.001** |
| | Ear pain | 1 | 1.4 | 10 | 71.4 | <0.001** |
| | No Neuro symptoms | 39 | 56.5 | 1 | 7.1 | 0.001** |
| | Any Neuro symptom | 30 | 43.5 | 13 | 92.9 | |
| Treated | Yes | 56 | 81.2 | 8 | 57.1 | 0.051** |
| | No | 13 | 18.8 | 6 | 42.9 | |
| Outcome | Survived | 47 | 68.1 | 2 | 14.3 | <0.001** |
| | Died | 20 | 29.0 | 12 | 85.7 | |
| | LAMA | 2 | 2.9 | 0 | 0.0 | |
| Lab tests | Protein in urine | 51 | 73.9 | 7 | 50.0 | 0.036** |
| | Blood in urine | 52 | 75.4 | 8 | 57.1 | 0.295** |
| Creatinine (umol/l) | <53 | 4 | 5.8 | 0 | 0.0 | |
| | 53–106 | 38 | 55.1 | 11 | 78.6 | 0.233** |
| | >106 | 27 | 39.1 | 3 | 21.4 | |
| Urea (mm/L) | <2.5 | 2 | 2.9 | 0 | 0.0 | |
| | 2.5–7.9 | 54 | 78.3 | 11 | 78.6 | 0.800** |
| | >7.9 | 13 | 18.8 | 3 | 21.4 | |
| AST (mmol/l) | <11 | 1 | 1.5 | 0 | 0.0 | |
| | 11–38 | 29 | 42.0 | 8 | 57.1 | 0.549** |
| | >38 | 39 | 56.5 | 6 | 42.9 | |
| ALT (mmol/l) | <10 | 6 | 8.7 | 0 | 0.0 | |
| | 10–47 | 47 | 68.1 | 11 | 78.6 | 0.493** |
| | >47 | 16 | 23.2 | 3 | 21.4 | |
| Platelet count | <150 | 27 | 39.1 | 6 | 42.9 | 0.795** |
| | 150–450 | 42 | 60.9 | 8 | 57.1 | |
| | Creatinine umol/LL [median (IQR)] | 105 (73–153) N = 55 | | 189 (84.5–293.5) N = 5 | | 0.309*** |

*(Continued)*

**Table 2.** (Continued)

| Characteristics | Outbreak 1 Dec, 2017 –Apr, 2018 | | Outbreak 2 Aug, 2018 –Dec, 2018 | | P value |
|---|---|---|---|---|---|
| | n | % | n | % | |
| AST (IU/L) [median (IQR)] | 54 (34–89) N = 55 | | 8 (60–142) N = 6 | | 0.118*** |
| Urine output mL/m²/24hrs [mean (SD)] | 1.9 (2.1) N = 69 | | 5.3 (3.9) N = 14 | | 0.001* |
| Platelet count x10⁹/l [mean (SD)] | 156.5 (82.0) N = 54 | | 87.8 (23.6) N = 6 | | 0.007* |

* Red eyes, subconjunctival haemorrhage; LAMA: Left Against Medical Advice; IQR: interquartile range; AST, Aspartate aminotransaminase; Neuro: Neurological

* T-test

** Chi squared test

*** Kruskal Wallis test.

## Discussion

Ebonyi State is one of the three high burdened states with frequent outbreaks of LF [39–41] (Fig 1). NCDC reported an epidemiological analysis and clinical aspects of the LF outbreak that occurred in Nigeria during January 1–May 6, 2018. Among the 3 states with the highest

**Table 3. Comparison of probable and confirmed cases for each outbreak.**

| Characteristics | | Outbreak 1 Dec, 2017 –Apr, 2018 | | Outbreak 2 Aug, 2018 –Dec, 2018 | | P value |
|---|---|---|---|---|---|---|
| | | Probable n (%) | Confirmed n (%) | Probable n (%) | Confirmed n (%) | |
| | Total | 6 | 63 | 7 | 7 | |
| Presenting symptoms | | | | | | |
| General | Fever | 6 (100.0) | 54 (85.1) | 7 (100.0) | 7 (100.0) | 0.783** |
| | Fatigue | 6 (100.0) | 52 (82.5) | 7 (100.0) | 7 (100.0) | 0.673** |
| | Bleeding | 5 (83.3) | 27 (42.9) | 6 (85.7) | 6 (85.7) | 0.091** |
| | Vomiting | 3 (50.0) | 29 (46.8) | 6 (85.7) | 75 (71.4) | 0.512** |
| | Diarrhoea | 3 (50.0) | 29 (46.0) | 5 (71.4) | 3 (42.9) | 0.895** |
| | Oedema | 2 (33.3) | 20 (31.8) | 2 (28.6) | 5 (71.4) | <0.001** |
| | Red eye* | 2 (33.3) | 18 (28.6) | 5 (71.4) | 6 (85.7) | <0.001** |
| Neurological | Altered consciousness | 3 (50.0) | 17 (27.0) | 7 (100.0) | 4 (57.1) | 0.012** |
| | Seizure | 1 (16.7) | 17 (27.0) | 7 (100.0) | 6 (85.7) | <0.001** |
| | Neck pain | 0 (0.0) | 8 (12.7) | 5 (71.4) | 5 (71.4) | <0.001** |
| | Hearing Impairment | 0 (0.0) | 4 (6.4) | 0 (0.0) | 4 (57.1) | <0.001** |
| | Ear pain | 0 (0.0) | 1 (1.6) | 5 (71.4) | 5 (71.4) | <0.001** |
| | No Neuro symptoms | 3 (50.0) | 36 (57.1) | 0 (0.0) | 1 (14.3) | 0.008** |
| | Any Neuro symptom | 3 (50.0) | 27 (42.9) | 7 (100.0) | 6 (85.7) | |
| Treated | Yes | 1 (16.7) | 54 (85.7) | 0 (0.0) | 7 (100.0) | <0.001** |
| | No | 5 (83.3) | 9 (14.3) | 7 (100.0) | 0 (100.0) | |
| Outcome | Survived | 0 (0.0) | 47 (77.1) | 0 (0.0) | 2 (28.6) | <0.001** |
| | Died | 6 (100.0) | 14 (23.0) | 7 (100.0) | 5 (71.4) | |
| | LAMA | 2 | 2.9 | 0 | 0.0 | |
| Epidemiological link established | Yes | 6 (100.0) | 44 (69.8) | 7 (100.) | 5 (71.4) | <0.001** |
| | No | 0 (0.0) | 19 (30.2) | 0 (0.0) | 1 (14.29) | |
| Time between onset of symptoms and admission | Median (IQR) | 12 (8–14.3) | 7 (4–11.5) | 14 (12–16) | 8 (7–11) | 0.024*** |

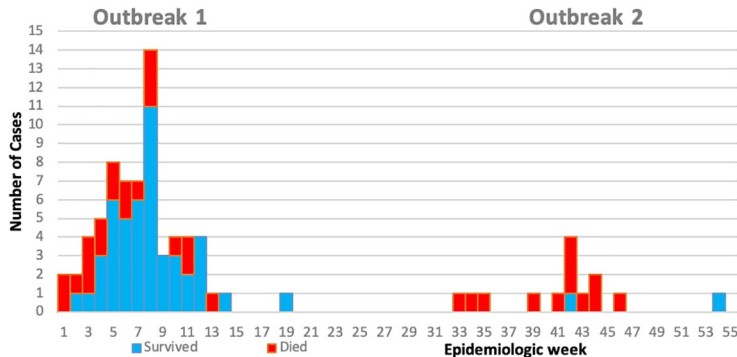

**Fig 4. Mortality distribution in the two outbreaks in Ebonyi State, 2017–18 of confirmed and probable cases.**

number of cases, the positive rates were 16.5% (Edo), 31.6% (Ondo), and 69.6% (Ebonyi). Ebonyi recorded the highest positive rate and among the 3 most affected states, CFR was 14.6% (Edo), 24.2% (Ondo), and 23.4% (Ebonyi) [41].

This study describes two Outbreaks of LF in Ebonyi State, Nigeria, that occurred at the beginning and later part of the same year but with different presentations. While the first outbreak followed the usual pattern of previous epidemics, the second outbreak demonstrated a completely different profile with early neurological manifestations, significant abnormal laboratory indices and a high case fatality rate. This latter presentation was entirely new in our environment, with no previous documentation in South Eastern Nigeria known to the authors after a thorough search of available literature. However, some previous studies on LF outside this environment have demonstrated early neurological signs with positive outcomes [42], poor outcomes [43], as well as late neurological manifestations [13].

Outbreak 2 involved farmers from rural areas almost exclusively while outbreak 1 cases were more likely to be urban residents and to be traders, students and health workers.

**Table 4. Logistic regression showing odds and adjusted odds for mortality for selected variables in the two outbreaks.**

| Variable | | Un-adjusted analysis | | | Model 1 | | | Model 2 : without neurological signs | | |
|---|---|---|---|---|---|---|---|---|---|---|
| | | OR | 95% CI | P value | AOR | 95% CI | P value | AOR | 95% CI | P value |
| Outbreak | 1 | 1 | | | 1 | | | 1 | | |
| | 2 | 14.1 | (2.9-68.9) | 0.001 | 6.6 | (0.8-52.0) | 0.074 | 10.1 | (1.4-73.0) | 0.007 |
| Days to presentation | 1 | 1 | | | 1 | | | 1 | | |
| | Per extra Day | 1.2 | (1.1-1.3) | 0.003 | 1.2 | (1.0-1.4) | 0.003 | 1.2 | (1.1-1.3) | 0.033 |
| Occupation | Civil servants | 1 | | | 1 | | | 1 | | |
| | Health worker | 1.7 | (0.3-9.8) | 0.572 | 2.7 | (0.2-30.0) | 0.426 | 2.7 | (0.2-11.3) | 0.426 |
| | Trader | 1.1 | (0.2-5.7) | 0.930 | 1.6 | (0.2-14.8) | 0.666 | 1.6 | (0.2-9.5) | 0.666 |
| | Farmer | 4.7 | (0.8-26.2) | 0.080 | 2.1 | (0.2-23.4) | 0.534 | 2.1 | (0.2-18.7) | 0.534 |
| | Student | 1.3 | (0.1-6.3) | 0.778 | 0.8 | (0.1-8.0) | 0.834 | 0.8 | (0.1-5.5) | 0.834 |
| | Children | | 0.8 | (0.1-6.3) | 0.814 | Empty | Empty | Empty | Empty | |
| Location | Rural | 1 | | | 1 | | | 1 | | |
| | Urban | 0.8 | (0.3-1.9) | 0.576 | 2.6 | (0.5-12.2) | 0.235 | 2.2 | (0.6-8.0) | 0.229 |
| Neurological signs | None | 1 | | | 1 | | | | | |
| | At least one | 24.1 | (6.2-92.3) | <0.01 | 14.1 | (3.1-65.0) | 0.001 | | | |

OR, odd ratio; AOR, adjusted odd ratio

Outbreak 2 cases presented at the Virology Centre with severe signs and symptoms, mostly neurological, as well as with more abnormal laboratory indices than those in outbreak 1. A possible explanation might be that outbreak 2 cases presented later in the course of their disease, however this was not demonstrated in the adjusted analysis on mortality, since outbreak 1 had a high number of farmers whose outcome was likely not better than cases in outbreak 2. Clearly, outbreak 2 cases were experiencing a different course of the disease. Whereas early commencement of treatment influenced the outcome of the cases in outbreak 1, high fatality rate was observed in outbreak 2 even amongst asymptomatic care givers that later developed symptoms while in the hospital presenting with neurological symptoms right from onset. This underscores the import of the second outbreak.

Lassa viruses are known for their diversity in Nigeria [31] and this study suggests the possibility of a more virulent strain with ominous implications given the high case fatality observed. Unfortunately, due to poor storage of the samples, phylogenetic analysis could not be carried out for the cases in outbreak 2 to determine if a different strain, variant or clade of the virus is evolving. This should be a priority for molecular research and the laboratory services. In future outbreaks, including biomolecular analysis would be crucial to further our understanding of epidemics and genetic characterisation of LF variants.

There are some strengths to this study. All the patients diagnosed with LF during the study period were included and all cases admitted in the Virology Centre had the same standardised care. There were no financial barriers to their receiving a full course of treatment unlike previous outbreaks when there were management challenges [44]. Currently the Virology Centre is well equipped with good human and material resources and has been certified by external bodies (NCDC, WHO). Surveillance and active case search by the Ebonyi State Ministry of Health for all symptomatic contacts enhanced follow up. The conduct and reporting of this study followed the international guidelines (STROBE—Strengthening the Reporting of Observational studies in Epidemiology) [45].

There were however some limitations. Most importantly, a substantial number of cases with strong epidemiologic links to LF died after reaching the Centre but before treatment could begin, thus limiting the amount of laboratory analysis that could be performed. Lumbar puncture is not routinely recommended for patients with neurological symptoms from LF due to the huge risk to staff of the procedure and the dangers to encephalopathic patients [46]. Therefore, it was not done but would have added more strength to the laboratory values. Also, biological sample storage was a big challenge as the integrity of the samples, particularly outbreak 2 were compromised limiting further genetic studies. LF is a rare disease, and the study had a relatively small sample size that impacted on its ability to carry out multivariate analysis for comparisons during analysis.

This study has several operational implications. Delays in presentation are a continuing challenge; patients have to be identified and transported early to the Virology Centre to receive the best chance of survival. This suggests a review of the referral system from primary health care centres to treatment centres for a more effective transportation network and surveillance system. In addition to genetic testing, there should also be clinical and laboratory studies focused on the characteristics of this uncommon presentation of LF. It also calls for newer, more rapid diagnostic tools for earlier detection, prompt management of cases, as well as improved and proper biospecimen management.

## Conclusion

In a similar time frame with a usual outbreak of LF in Ebonyi State, Nigeria, this study revealed an unusual form of presentation with a different profile of signs and symptoms and a high

mortality rate in the rural setting of our locality. It is a cause for great public health concern and must be recognised and investigated more fully in future outbreaks as this could be a different variant of LF.

## Supporting information

**S1 Table. Clinical Stages of Severe LF.**
(DOCX)

**S2 Table. NCDC Case Definitions.**
(DOCX)

**S3 Table. Ribavirin Regimen (McCormick Regimen).**
(DOCX)

**S4 Table. Clinical symptoms significantly associated with mortality in the two outbreaks.**
(DOCX)

## Acknowledgments

This research was conducted through the Structured Operational Research and Training Initiative (SORT IT), a global partnership led by the Special Programme for Research and Training in Tropical Diseases at the World Health Organization (WHO/TDR). The model is based on a course developed jointly by the International Union Against Tuberculosis and Lung Disease (The Union) and Medécins sans Frontières (MSF/Doctors Without Borders). The specific SORT IT programme which resulted in this publication was jointly organised, implemented and mentored by the Centre for Operational Research, The Union, Paris, France; MSF-Luxembourg (MSF LuxOR); MSF-Belgium (MSF-OCB); the University of Bergen, Norway and the London School of Hygiene and Tropical Medicine, London, UK.

Special gratitude to all the staff of the Virology Center of AEFUTHA who were involved in the care of the patients in this study. Many thanks to the Management of Alex Ekwueme Federal University Teaching Hospital, Ebonyi State Ministry of health, Nigeria Center for Disease Control, MSF—Nigeria, and WHO for all their support in the Outbreak control.

## Author Contributions

**Conceptualization:** Nneka M. Chika-Igwenyi, Uche S. Unigwe, Nnennaya A. Ajayi.

**Data curation:** Nneka M. Chika-Igwenyi.

**Formal analysis:** Rebecca E. Harrison, Julita Gil-Cuesta, Obumneme Ezeanosike, Jacques K. Tshiang.

**Funding acquisition:** Anthony Reid.

**Investigation:** Nneka M. Chika-Igwenyi, Damian U. Nwidi, Uchenna Anebonam.

**Methodology:** Nneka M. Chika-Igwenyi, Christina Psarra, Maria Gulamhusein, Uche S. Unigwe, Nnennaya A. Ajayi, Anthony Reid.

**Project administration:** Nneka M. Chika-Igwenyi, Nnennaya A. Ajayi.

**Resources:** Emeka O. Onwe, Robinson C. Onoh, Ugochukwu U. Nnadozie, Chiedozie K. Ojide, Azuka S. Adeke, Collins N. Ugwu.

**Software:** Rebecca E. Harrison, Obumneme Ezeanosike.

**Supervision:** Nnennaya A. Ajayi.

**Validation:** Uche S. Unigwe, Emeka Sampson.

**Visualization:** Nneka M. Chika-Igwenyi.

**Writing – original draft:** Nneka M. Chika-Igwenyi, Maria Gulamhusein, Anthony Reid.

**Writing – review & editing:** Nneka M. Chika-Igwenyi, Christina Psarra, Julita Gil-Cuesta, Robinson C. Onoh, Azuka S. Adeke, Jacob Maikere, Anthony Reid.

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
