## [Decision Letter · Decision Letter 0]

2 Oct 2020

Dear Dr. Chika-Igwenyi,

Thank you very much for submitting your manuscript "EARLY ONSET OF NEUROLOGICAL FEATURES DIFFERENTIATES TWO OUTBREAKS OF LASSA FEVER IN EBONYI STATE, NIGERIA DURING 2017-2018" for consideration at PLOS Neglected Tropical Diseases. As with all papers reviewed by the journal, your manuscript was reviewed by members of the editorial board and by several independent reviewers. In light of the reviews (below this email), we would like to invite the resubmission of a revised version that takes into account the reviewers' comments. 

We cannot make any decision about publication until we have seen the revised manuscript and your response to the reviewers' comments. Your revised manuscript is also likely to be sent to reviewers for further evaluation.

Sincerely,

Andrea Marzi

Deputy Editor

Reviewer's Responses to Questions

**Key Review Criteria Required for Acceptance?**

**Methods**

-Are the objectives of the study clearly articulated with a clear testable hypothesis stated?

-Is the study design appropriate to address the stated objectives?

-Is the population clearly described and appropriate for the hypothesis being tested?

-Is the sample size sufficient to ensure adequate power to address the hypothesis being tested?

-Were correct statistical analysis used to support conclusions?

-Are there concerns about ethical or regulatory requirements being met?

Reviewer #1: Methods are mostly acceptable but require some clarifications. See general comments below.

Reviewer #2: No concerns.

Reviewer #3: -Are the objectives of the study clearly articulated with a clear testable hypothesis stated? Yes

-Is the study design appropriate to address the stated objectives? Yes, limitations largely recognized.

-Is the population clearly described and appropriate for the hypothesis being tested? Given it is a rare disease and the study is localized to a particular geography, yes.

-Is the sample size sufficient to ensure adequate power to address the hypothesis being tested? For some, but not all analysis approaches, but limitation was recognized.

-Were correct statistical analysis used to support conclusions? As above, yet.

-Are there concerns about ethical or regulatory requirements being met? No, the study appears to meet ethical considerations specific to the organizations involved and no obvious ethical/regulatory fouls are recognized.

**Results**

-Does the analysis presented match the analysis plan?

-Are the results clearly and completely presented?

-Are the figures (Tables, Images) of sufficient quality for clarity?

Reviewer #1: Suitable, though the stats need to be redone to separate confirmed and probable cases. See below

Reviewer #2: Improvement needed.

Reviewer #3: -Does the analysis presented match the analysis plan? By and large, yes.

-Are the results clearly and completely presented? Yes

-Are the figures (Tables, Images) of sufficient quality for clarity? Yes

**Conclusions**

-Are the conclusions supported by the data presented?

-Are the limitations of analysis clearly described?

-Do the authors discuss how these data can be helpful to advance our understanding of the topic under study?

-Is public health relevance addressed?

Reviewer #1: Conclusion seem appropriate

Reviewer #2: No concerns.

Reviewer #3: -Are the conclusions supported by the data presented? See comments below as there appear to be some other important considerations related to their findings perhaps not otherwise considered.

-Are the limitations of analysis clearly described? Many are.

-Do the authors discuss how these data can be helpful to advance our understanding of the topic under study? Yes

-Is public health relevance addressed? Yes

**Editorial and Data Presentation Modifications?**

Reviewer #1: (No Response)

Reviewer #2: See general comments.

Reviewer #3: Line 73: These are fairly old references that made a lot of assumptions of the asymptomatic burden within the population based on seroprevalence studies at the time. I'm not sure they are entirely accurate and would recommend the authors mute or blunt this statement.

Line 84: Hospitalized case CFR in Nigeria or Sierra Leone? They are recognized to be different by those working in the field, likely due to differential standard of care in each setting. Perhaps viral genotypes as well, but not enough data to confirm.

Line 119: AEFUTHA first use, spell out acronym.

Line 200: Readers would benefit by understanding a better definition of what a "trader" does. Is this a highly transient merchant that spends time in both rural and developed areas?

Line263-264: Given the high OR of farmers from either outbreak, readers would benefit from understanding if any differences in time to presentation between outbreaks (i.e. are farmers from rural settings showing up later to the hospital than farmers from more urban areas?). It is well established that persons from more rural settings are less likely to seek medical attention and likely present due to unfortunate issues associated with limited transportation access, social stigmas, or even fears of treatment (ie risks of going to hospital). 

Would also be wise to recognize the role of increased/improved surveillance or even public health outreach campaigns in either outbreak setting? Are there differences in the public education programs in these two settings?

Line 301: Its possible that for the area of Ebonyi State, that the neurological presentation timelines associated with increased mortality are unusual; but by enlarge,these features are common to severe LF in humans. I would recommend muting this statement or making it very specific to the geography of this study. 

A map of the two outbreak sites would help the reader understand the geographic separation between the outbreaks and support the potential for different viral isolates to be be responsible for these two outbreaks.

**Summary and General Comments**

Reviewer #1: The article by Chika-Igwenyi et al. describes clinical aspects of two separate Lassa fever outbreaks in a defined geographical area of Nigeria. The author’s main finding is an increased presence of neurological symptoms during the second outbreak when compared to the first outbreak earlier in the year. The manuscript is well suited for PLoS neglected tropical diseases and presents some interesting data that will be highly useful in the field. However, there are some points that need clarification to improve the interpretation of the data. 

1. Given the high genetic diversity of Lassa virus in Nigeria and with at least three known clades documented in this country it is important to provide some genetic analysis on these cases to minimally determine the infecting clade. The increases in neurological manifestations may be due to a differing clade of virus, which in itself would be highly interesting. 

2. I suggest adding a map of Nigeria, highlighting Ebonyi state as well as other regions in Nigeria that were experiencing Lassa fever outbreaks during this time period.

3. The definition of a confirmed and probable case must be included in the text.

4. Please describe the molecular diagnostics utilized in identifying LF cases. One vs two target, what gene/region was amplified, etc.

5. Did other concurrent outbreaks of LF in Nigeria also observe increased neurological manifestations? 

6. The benefit of ribavirin in treating LF cases is highly controversial, possibly due in part to a lack of publications outlining its use. Please discuss any benefits of ribavirin treatment observed in patients from this study

7. The data tables and associated statistics should be re-done to differentiate probable and confirmed cases. Alternatively, you could exclude the probable cases from these tables since you did not confirm them as Lassa positive / LF cases. The statistics are misleading when everything is grouped together.

8. Please define the abnormal laboratory tests. Other laboratory tests run should also be mentioned, even if they do not differ between the two outbreaks 

Minor comments: 

1. Once defined after first use, please use LF and LASV throughout the manuscript

2. Outbreak does not need to be capitalized

3. What is a mini morgue?

4. Please specify the statistical tests used in the table footnote so that readers do not have to search for it.

Reviewer #2: Chika-Igwenyi and co-workers describe the epidemiology, socio-demographics, clinical characteristics, and outcomes of patients seen during two Lassa fever outbreaks in Ebonyi State, Nigeria between December 2017 and 39 December 2018. The results are important, but a few improvements in data presentation are needed. 

Table 3 shows much of the same data as Table 2 can be eliminated or placed in Supplemental Material.

The normal range for clinical parameters, such as creatinine and AST, should be provided in Tables 2 and 3. Elevated creatinine and AST have been observed in other Lassa fever clinical cohorts in Nigeria and Sierra Leone. AST levels in the current study appear closer to the normal range. These prior studies and differences/similarities should be discussed. 

Headers should be added to Table 4 to indicate the different analyses.

Reviewer #3: Dr. Chika-Igwenyi and colleagues provide an epidemiological analysis of two outbreaks in Ebonyi State, Nigeria from 2018-2019. The work is strictly epidemiological with support of clinical information based on disease severity and associated symptoms of disease. The central claim appears to be that between the two outbreaks, different disease manifestations surfaced that were associated with differential mortality outcomes. Unfortunately there is no genetic information of the virus isolates associated with either outbreak, thus making interpretation of the possibility of two different viruses variants being responsible for the difference in outcome. LASV infection classically is associated with late onset neurological signs of which chances for poor outcome are increased if no supportive therapy or pharmaceutical interventions are started. The work might also be assisted by reporting CT values (or genomic equivalent values if possible) of the real-time RT-PCR to get a better idea of what state the persons were in upon admission to the hospital for treatment. As mentioned elsewhere in this review, it is not uncommon for persons in rural setting to delay seeking treatment for mainly socio-economic reasons, so it was not at all surprising to see more severe disease manifestations in that population. But I think more data related to disease state and viral burden, if not the phylogenetic analysis would be necessary to confidently ascertain these outbreaks were indeed caused by different.

I understand that many of my comments may come as difficult, but I hope the authors will not be discouraged as they are only intended to strengthen the work as I believe this work is very important and would certainly benefit the LASV research community the opportunity to see their work; however, some needed refinements are needed prior to general release.

PLOS authors have the option to publish the peer review history of their article (what does this mean?). If published, this will include your full peer review and any attached files.

Reviewer #1: No

Reviewer #2: Yes: Robert F Garry

Reviewer #3: No
---

## [Editor Report · Decision Letter 1]

22 Jan 2021

Dear Dr. Chika-Igwenyi,

We are pleased to inform you that your manuscript 'EARLY ONSET OF NEUROLOGICAL FEATURES DIFFERENTIATES TWO OUTBREAKS OF LASSA FEVER IN EBONYI STATE, NIGERIA DURING 2017-2018' has been provisionally accepted for publication in PLOS Neglected Tropical Diseases.

Best regards,

Andrea Marzi

Deputy Editor

---

## [Editor Report · Acceptance letter]

18 Feb 2021

Dear Dr. Chika-Igwenyi,

We are delighted to inform you that your manuscript, "EARLY ONSET OF NEUROLOGICAL FEATURES DIFFERENTIATES TWO OUTBREAKS OF LASSA FEVER IN EBONYI STATE, NIGERIA DURING 2017-2018," has been formally accepted for publication in PLOS Neglected Tropical Diseases.

Best regards,

Shaden Kamhawi

co-Editor-in-Chief

Paul Brindley

co-Editor-in-Chief
